# Hair Growth Effect and the Mechanisms of *Rosa rugosa* Extract in DHT-Induced Alopecia Mice Model

**DOI:** 10.3390/ijms252111362

**Published:** 2024-10-22

**Authors:** Ha-Rim Kim, Jung Up Park, Seung-Hyeon Lee, Jae Young Park, Wonwoo Lee, Kyung-Min Choi, Seon-Young Kim, Mi Hee Park

**Affiliations:** 1Jeonju AgroBio-Materials Institute, Wonjangdong-gil 111-27, Deokjin-gu, Jeonju-si 54810, Jeonbuk-do, Republic of Korea; poshrim@jami.re.kr (H.-R.K.); sh94@jami.re.kr (S.-H.L.); jjay1205@jami.re.kr (J.Y.P.); 2Division of Practical Research, Honam National Institute of Biological Resources, 99, Gohadoan-gil, Mokpo-si 58762, Jeollanam-do, Republic of Korea; pju2560@hnibr.re.kr (J.U.P.); 21cow@hnibr.re.kr (W.L.); kyungmc0111@hnibr.re.kr (K.-M.C.); 3Advanced Research Center for Island Wildlife Biomaterials, Honam National Institute of Biological Resources, 99, Gohadoan-gil, Mokpo-si 58762, Jeollanam-do, Republic of Korea

**Keywords:** *Rosa rugosa* extract, hair growth, dihydrotestosterone, alopecia, mice model

## Abstract

*Rosa rugosa* is a medicinal plant known for its potential anti-inflammatory, antioxidant, anti-cancer, and antimicrobial benefits. The pharmacological effects of *Rosa rugosa* extract on hair loss have not yet been documented. This research sought to assess the inhibitory effects and mechanisms of action of *Rosa rugosa* water extract (RWE) in a mouse model of dihydrotestosterone (DHT)-induced alopecia. The study was conducted using C57BL/6 mice, which were assigned to five groups: control, DHT-treated, *Rosa rugosa* water extract (RWE) at doses of 25 mg/kg and 100 mg/kg body weight, and bicalutamide-treated. To induce hair loss, dihydrotestosterone (1 mg/day per body weight) was administered via intraperitoneal injections, and dorsal hair removal was timed to align with the telogen phase. Each group received oral treatments for a period of 23 days. In this study, we assessed hair growth activity, examined histological changes, and performed immunoblot analysis. We noted improvements in hair length and thickness. Additionally, the protein expression of growth factors associated with hair growth, including vascular endothelial growth factor (VEGF), epidermal growth factor (EGF), and insulin-like growth factor-1 (IGF-1), showed significant increases in the group treated with RWE. Additionally, treatment with RWE suppressed the protein expression of hair growth inhibitory factors, including dickkopf WNT signaling pathway inhibitor 1 (DKK1) and interleukin (IL)-6. Moreover, hair growth regulatory pathway related factors, including ERK, AKT, and GSK-3β, were activated. These findings indicate that RWE could serve as a promising natural therapy for preventing hair loss by enhancing the production of factors that promote hair growth while inhibiting those that suppress it.

## 1. Introduction

The increasing incidence of alopecia, which frequently causes significant psychological distress, has intensified efforts to find effective treatments for hair loss [1]. As individuals grow older, hair loss and thinning can decrease hair density and negatively impact their quality of life [2]. Exposure to external stressors and the effects of aging can lead to accumulated oxidative stress, which interferes with normal hair cycles and results in a decline in stem cell function, ultimately contributing to ongoing hair loss [3]. Hair density is largely affected by the control of the hair growth cycle, which is regulated by a range of hormones and growth factors [4]. A crucial factor is androgen-related hormonal imbalance, often associated with androgenetic alopecia, which results in the gradual shrinking of hair follicles without causing scarring and reduces the duration of the anagen phase. This condition affects both genetically predisposed men and women [5]. Various growth factors, such as insulin-like growth factor-1 (IGF-1), vascular endothelial growth factor (VEGF), epidermal growth factor (EGF), fibroblast growth factor (FGF), and keratinocyte growth factor, play essential roles in progressing different stages of the hair cycle and are pivotal for promoting hair growth [6,7]. Hair loss is commonly associated with factors such as aging, genetic factors, hormonal changes, and insufficient nutrition. Recent research indicates that dietary supplements rich in essential nutrients might increase the number of hair follicles in women suffering from telogen effluvium, suggesting these supplements could potentially mitigate and delay hair loss [8].

Given the substantial impact of hair loss on social interactions and mental health, there is an increasing focus on both prevention and treatment. The three main types of FDA-approved treatments for hair loss are topical minoxidil, oral finasteride, and low-level laser therapy [9]. Minoxidil can lead to side effects like skin irritation and contact dermatitis [10]. Finasteride is frequently prescribed for male pattern baldness, but it is generally unsuitable for female patients [11]. Consequently, there is a pressing need for more effective and safer treatment options. With the growing interest in products that promote hair growth, numerous nutraceuticals are being investigated as potential alternative therapies for hair loss and thinning [12]. Recent studies suggest that natural products may be effective in delaying hair loss.

Various vitamin supplements and natural products, including saw palmetto, caffeine, melatonin, marine extracts, rosemary oil, procyanidin, pumpkin seed oil, and cannabidiol oil, have been investigated for their potential benefits in treating androgenetic alopecia and telogen effluvium [13,14]. Patients with non-cicatricial alopecias have been found to have lower serum levels of vitamin D and ferritin compared to individuals without these conditions [15,16]. Micronutrients may significantly influence alopecia by impacting hair follicle development and the function of immune cells. Additionally, changes in nutrition are linked to telogen effluvium, which can affect the effectiveness of treatments for female pattern hair loss [17]. Also, biotin and zinc supplementation have been used for the treatment of hair loss; however, there is currently no evidence to suggest that supplementing with these substances leads to improvements in hair loss [18]. Although numerous natural plant products are available to address hair loss, a definitive cure has not yet been found.

*Rosa rugosa*, commonly known as rugosa rose and a member of the Rosaceae family, originates from East Asia and is now widely grown around the world [19]. Known for its fragrant flowers, *Rosa rugosa* is often used to produce rose hip oil, which is abundant in aromatic, sweet, and biologically active volatile compounds [20]. *Rosa rugosa* has been employed to address various health issues, such as inflammation, stomach discomfort, menstrual irregularities, diarrhea, pain, diabetes, bleeding disorders, colitis, hemoptysis, strokes, paralysis, dysmenorrhea, colds, flu-like symptoms, infections, and fevers [21]. Recently, rose hips have garnered interest in cosmetic treatments due to their rich vitamin C content [22]. Studies on the fruit of *Rosa rugosa*, or beach rose, have investigated its antioxidant properties and its composition of carotenoids, polyphenols, flavonoids, vitamins, terpenoids, steroids, and tocopherols [23]. *Rosa rugosa* powder exhibits anti-inflammatory properties that may be beneficial for osteoarthritis [24]. *Rosa rugosa*, obtained from its flesh and seeds, influences skin wrinkles, cell aging, and the overall aging process [25]. *Rosa rugosa* powder is a crucial and potent ingredient that prevents aging and supports cell longevity [22]. Traditionally, extracts from the flowers and hips of *Rosa rugosa* have been highly regarded in Asian cuisine, cosmetics, aromatherapy, and medicine [26]. The diverse effects observed are likely due to the variety of phytochemicals present in *Rosa rugosa* extracts, including flavonoids, phenylpropanoids, bisabolanes, acoranes, tannins, fatty acids, and terpenoids [27]. However, the effects of *Rosa rugosa* extract on hair loss have not been reported yet.

In this research, we investigated for the first time the effects of a new oral RWE nutraceutical formulation on hair growth in C57BL/6 mice. The study involved administering the RWE formulation at two distinct concentrations over a span of 3 weeks to evaluate its impact on hair growth. We examined the expression levels of genes and proteins associated with hair growth in the skin tissues of the C57BL/6 mice that received the RWE treatment.

## 2. Results

### 2.1. Effect of RWE on Hair Growth in C57BL/6 Mice

To investigate the effects of RWE on hair growth in C57BL/6 mice, we acclimatized 5-week-old C57BL/6 mice for 1 week and intraperitoneally injected DHT (1 mg/kg b.w.). After 4 days, the telogen phase was initiated by removing the hair from the mouse. RWE (25 mg/kg b.w., 100 mg/kg b.w.) and bicalutamide, an androgen receptor antagonist drug, were orally administered. As shown in Figure 1, at 11 days, we observed that RWE (100 mg/kg b.w.) promoted hair growth more prominently than the control or DHT group. Regrowth of hair was delayed significantly in the DHT-stimulated group, while the RWE (100 mg/kg b.w.) group showed almost similar regrowth to that of the control group. These results suggest that administration of RWE promotes hair follicle regrowth in DHT-induced hair growth retardation in mice.

### 2.2. Impact of RWE on the Protein Expression of Hair Growth Factors in the Skin Tissue of C57BL/6 Mice

To examine the impact of RWE on the protein expression of growth factors, we utilized Western blotting or ELISA using skin tissue and serum. We assessed the levels of VEGF and IGF-1, two crucial growth factors that promote hair growth. Both VEGF and IGF-1 protein expression showed a marked increase in the groups receiving low and high doses of RWE compared to the negative control group (Figure 2A,B). Furthermore, serum EGF levels were measured and found to be increased in the RWE treatment group (Figure 2C).

### 2.3. Impact of RWE on the Protein Expression of Hair Growth Inhibitory Factors in the Skin Tissue of C57BL/6 Mice

We examined the impact of RWE on the protein expression of growth inhibitory factors, including DKK-1 and IL-6, in the skin tissues. The results showed that the expression of these genes reduced in the group treated with RWE 100 mg/kg (Figure 3A). The expression of the IL-6 gene was statistically significantly reduced (Figure 3B).

### 2.4. RWE Improves Hair Follicle Regeneration in C57BL/6 Mice

The reduction in the anagen phase is a key factor in terminal hair loss. To examine the development of hair follicles across the hair cycle, we performed hematoxylin–eosin staining (Figure 4A). By referring to histological studies, we identified hair follicles in the anagen phase. In the negative control group, approximately 41.8% of hair follicles were in the anagen phase, while the RWE-administered groups at doses of 25 mg/kg and 100 mg/kg body weight exhibited 66.4% and 70.6% of hair follicles in the anagen phase, respectively, as shown in Figure 4C. An elevation in both the quantity and dimensions of hair follicles is often considered a sign of the shift from the telogen to anagen phases of hair growth. The size and number of hair follicles are especially conspicuous in the deep subcutaneous layers during the anagen phase. Furthermore, in longitudinal and transverse sections, the RWE-administered group showed a higher number of hair follicles compared to the DHT-treated group. As seen in Figure 4B, the RWE-administered group had a greater number of hair follicles than the DHT-treated group.

### 2.5. Effect of RWE on Hair Thickness of C57BL/6 Mice

To assess hair thickness, we analyzed it using a scanning electron microscope (SEM) with an 80× lens (Figure 5A and Figure 5C, respectively). We also took a picture of the hair in each sample (Figure 5B). The DHT-treated group exhibited significantly thinner hair, whereas the hair thickness in the RWE-administered group resembled that of the control group.

### 2.6. Impact of RWE on the Protein Expression Related with Hair Growth Regulatory Pathways in Skin Tissue of C57BL/6 Mice

To examine the impact of RWE on the protein expression of growth factor regulatory pathways, we utilized Western blotting using skin tissue. We assessed the levels of p-GSK-3β, pERK, and p-AKT that promote hair growth. We showed a marked increase in p-GSK-3β, pERK, and p-AKT in the groups receiving low and high doses of RWE compared to the negative control group (Figure 6A,B).

## 3. Discussion

In this study, we demonstrated for the first time that extract from *Rosa rugosa* is effective against DHT-induced alopecia. Our findings showed that oral administration of RWE stimulated hair regrowth and induced changes in hair thickness in DHT-induced mice. Furthermore, RWE increases the expression genes regulating hair growth such as VEGF, IGF-1 and EGF, and suppresses the expression of genes associated with hair loss, including DKK-1 and IL-6 induced by DHT.

The hair growth cycle is divided into three separate phases: anagen, catagen, and telogen [28]. Physiologically, notable alterations in skin blood flow play a role in the cycling of hair follicles, which includes the resting phase (telogen), the growth phase (anagen), and the regression phase (catagen). Various growth factors, such as VEGF, IGF-1, and EGF control the hair growth cycle [28]. Disruptions in these growth factors can result in hair loss. Studies have emphasized the important role of VEGF as a central regulator of both normal and abnormal blood vessel formation during skin development [29]. During the anagen phase, epithelial hair bulbs exhibit active angiogenic properties. In contrast, the catagen phase involves the regression of hair follicles, leading to the degeneration of capillary loops in the dermal papilla, which probably decreases blood flow to the hair matrix cells [30]. Notably, conditions such as androgenetic alopecia are often linked to decreased blood supply to the hair follicles [30]. In the anagen phase, the hair follicle depends largely on VEGF to maintain sufficient vascularization around it [31]. A previous study found that transgenic mice with VEGF overexpressed in hair follicle keratinocytes exhibited a significant increase in perifollicular vascularization throughout the hair growth cycle. These mice also showed faster hair regrowth after depilation and larger hair follicles with increased hair shaft diameter. This suggests that VEGF plays a vital role in regulating perifollicular vascularization during hair growth [30]. In addition, GSK-3β integrates extracellular and intracellular signals and regulates the expression of various growth factors such as VEGF. Based on this research, we observed that VEGF expression was elevated with RWE treatment in DHT-induced mice. Additionally, we identified key growth factors such as EGF and IGF-1. EGF plays a critical role in hair follicle growth and development by stimulating the proliferation of mesenchymal stem cells derived from hair follicles through the activation of the AKT and EGFR/ERK signaling pathways [32]. In this study, we confirmed that RWE activates AKT/ERK pathways, moreover, it activates GSK-3β phosphorylation. GSK-3β acts as a downstream effector of the PI3K/AKT pathway and negatively regulates the β-catenin, a key factor for morphogenetic formation of the hair follicles, and therefore, phosphorylation of GSK-3β is one of the critical reaction steps for hair follicle growth [33]. RWE could increase the phosphorylation of ERK and AKT, and then GSK-3β phosphorylation is enhanced by p-AKT activation. The phosphorylated GSK-3β loses the ability to activate β-catenin phosphorylation, leading to the accumulation of β-catenin in the nucleus to upregulate β-catenin-dependent transcription through the Wnt/β-catenin pathway [34]. Eventually, RWE promotes hair follicle growth. EGF supports the maintenance and renewal of the epidermal layer and affects the migration of stromal cells, but it does not impact the elongation of the outer root sheath towards the hair shaft [35]. In this study, RWE treatment promoted EGF expression in skin tissues. Additionally, IGF-1 plays a key role in regulating various cellular functions and biological processes [36]. Recent studies show that IGF-1 influences the hair growth cycle by impacting follicular proliferation, differentiation, and tissue remodeling [37]. IGF-1 supports hair growth and is crucial for the hair follicle cycle and development in transgenic mice, with its expression being associated with androgenetic alopecia [38]. Moreover, IGF-1 plays a role in managing the shift from the anagen phase to the catagen phase of the hair follicle cycle [39]. Previous research has shown that IGF-1 regulates cell apoptosis and promotes cell proliferation, as observed in its effects within the dermal papilla [40]. In this study, we also found that the expression of these growth factors, including EGF and IGF-1, increased by treatment of RWE compared with the DHT-treated group.

Androgenetic alopecia is the most prevalent form of hair loss, characterized by disrupted hair growth cycles [9]. The excessive production of DHT by 5-α-reductase in human hair dermal papilla cells is a primary factor in the development of androgenetic alopecia [41]. DHT binds to androgen receptors with an affinity that is approximately 5 to 10 times higher than that of testosterone [42]. Androgenetic receptor-bound DHT increases the expression of hair growth-inhibiting factors like DKK-1, IL-6, and transforming growth factor β (TGF-β), which in turn promotes hair follicle regression [43]. We also confirmed these mechanisms. As shown in our results, the expression of DKK-1 and IL-6 was induced by DHT; however, it decreased by the administration of RWE.

Hair loss is highly prevalent, but current treatments are limited and often come with significant side effects. As a result, there is significant interest in investigating natural products that support hair growth. Many of these natural products possess antioxidant properties, which have been shown to be effective in treating hair loss. Treatment with resveratrol in human hair follicles stimulated hair shaft growth and delayed the progression of the catagen phase [44]. The antioxidant compound arctiin, derived from *Arctium lappa*, has been demonstrated to shield dermal papilla cells from oxidative damage [45]. A clinical trial assessing a topical nutritional supplement comprising omega-3 and omega-6 fatty acids from fish and blackcurrant seed oils, along with vitamin C, vitamin E, and lycopene, demonstrated that the supplement positively impacted patients’ hair by decreasing the number of telogen and miniaturized hair follicles, which led to a significant increase in hair density [46]. A novel topical lotion, featuring a blend of two polyphenol components including dihydroquercetin-glucoside and epigallocatechin gallate-glucoside, has been employed in the treatment of androgenetic alopecia [47]. Micronutrients, particularly vitamins and minerals, are essential components of the hair follicle cycle and play a crucial role in managing hair loss [48]. Previous phytochemical studies on *Rosa rugosa* have revealed that its flowers are abundant in phenolic compounds such as flavonoids, tannins, and anthocyanins, which are recognized for their antioxidant effects [20]. Reports indicate that an extract from *Rosa rugosa* exhibits anti-inflammatory effects, anti-aging, whitening, and moisturizing properties [49]. So, antioxidant components in *Rosa rugosa* may have therapeutic effect in hair loss. We also analyzed RWE using UPLC (Appendix A) and Q-TOF-MS (Appendix A). We found that a variety of compounds were detected in the RWE of the chromatogram data (Appendix A). Additionally, 16 substances, including antioxidants, were found in RWE (Appendix A). However, further research is needed to identify important key components.

To test whether RWE affects body weight and causes liver damage, body weight and the levels of aspartate aminotransferase (AST) and alanine aminotransferase (ALT) were tested. We showed that the RWE has no effect on the body weight (Appendix A). Moreover, ALT (Appendix A) and AST (Appendix A) serum levels did not significantly change in the RWE treated group.

In summary, our findings indicate that RWE could serve as a promising natural treatment for preventing hair loss by enhancing the expression of factors associated with hair growth. Particularly, treatment with RWE led to improvements in both hair growth and thickness, as well as elevated protein expression levels of growth factors such as VEGF, EGF, and IGF-1. Collectively, these observations suggest that RWE could be a promising functional food material to improve hair growth and overall hair health or could be used as a drug for hair loss. Following the positive results from our preclinical study, we are in the process of developing a clinical trial to assess the effects of RWE on hair growth in future research.

## 4. Materials and Methods

### 4.1. Sample Preparation

The biological resources used in this research were distributed from the BOBIC (Bank of Bioresources from Island and Coast, Mokpo-si, Jeollanam-do, Republic of Korea). Dried *Rosa rugosa* flower buds were extracted once in distilled water at 100 °C for 4 h. The filtered crude extracts were freeze-dried in a vacuum dryer (770 mmHg).

### 4.2. Animals

Male C57BL/6J mice (5 weeks old, 20–24 g) were procured from Damul Science (Daejeon, Republic of Korea). The mice were randomly grouped with three per cage and kept under a 12 h light/dark cycle at a temperature of 22 ± 2 °C and 55 ± 5% humidity. They were allowed a 1-week acclimation period with unrestricted access to food and water. The Animal Care Committee of the Jeonju AgroBio-Materials Institute reviewed and approved the experimental procedures for this study, ensuring compliance with their guidelines throughout (JAMI IACUC 2023009).

### 4.3. Induction of DHT-Induced Hair Loss in Mice

DHT (1 mg/kg) was administered intraperitoneally for 4 days to induce hair loss. On day 0, hair removal was carried out using an animal clipper and razor, and was further assisted with hair removal cream. Mice were divided into seven groups of ten, with treatments administered orally for 23 days from the day of hair removal. The groups were as follows (Table 1). Subsequently, hair growth measurements were recorded through photographs taken while the animals were under anesthesia.

### 4.4. Histological Analysis

To evaluate hair regrowth, skin samples obtained post-depilation were fixed in 4% formaldehyde and embedded in paraffin blocks for further analysis. The paraffin blocks was cut into 5 μm thick sections, either longitudinally or horizontally. The sections underwent deparaffinization and were stained with hematoxylin and eosin (H&E), then examined using a light microscope. On day 23, all mice were sacrificed to obtain skin tissue samples, and analysis of the hair cycle was conducted with reference to previous studies [50,51]. Body weight gain and liver weight were measured (Appendix A).

### 4.5. Hair Thickness Analysis

A minimum of 200 hairs per mouse were gathered for each experimental group, as described [52]. Hair thickness was assessed using a scanning electron microscope (TM4000plus, Hitachi High-Technologies, Tokyo, Japan) with a magnification of ×80. Hair thickness was measured using Image J software version 1.54.

### 4.6. Analysis of Hair Growth Stimulatory and Inbibitory Factors

Skin tissue was lysed using ice-cold RIPA buffer and centrifuged at 10,000× *g* for 10 min. The supernatant was subjected to SDS-PAGE, and proteins were transferred to PVDF membranes. These membranes were blocked with 5% skim milk and incubated overnight with primary antibodies (anti-VEGF (1:1000, Santa Cruz, CA, USA), anti-IGF-1 (1:1000, Abcam, Cambridge, UK), anti-phospho-GSK3β (1:1000, Cell Signaling, Danvers, MA, USA), anti-GSK3β (1:1000, Cell Signaling, Danvers, MA, USA), anti-phospho-ERK (1:2500, Cell Signaling, Danvers, MA, USA), anti-ERK (1:2500, Cell Signaling, Danvers, MA, USA), anti-phospho-AKT (1:1000, Cell Signaling, Danvers, MA, USA), anti-AKT (1:1000, Cell Signaling, Danvers, MA, USA), anti-DKK-1 (1:1000, Santa Cruz, CA, USA), anti-IL-6 (1:1000, Invitrogen, Waltham, MA, USA)). Subsequently, HRP-conjugated secondary antibodies (anti-rabbit IgG (1:5000, Cell Signaling), anti-mouse IgG (1:5000, Cell Signaling)) were applied. Protein expression was detected using ECL solution and visualized with an Amersham imager 600 (GE Healthcare, Buckinghamshire, UK).

The concentration of EGF (epidermal growth factor) in serum was measured using a commercial EGF ELISA kit (Abcam, Cambridge, UK). The assay was performed according to the manufacturer’s protocol. Briefly, each serum of sample and antibody cocktail was added to the ELISA plate. The plate was incubated according to the manufacturer’s instructions. After the washing step, TMB substrate was added to develop the color reaction. Then, the reaction was stopped by the addition of stop solution. The absorbance was measured at 450 nm using a microplate reader.

### 4.7. Hepatotoxicity Indicators Analysis

Blood was drawn from the inferior vena cava and centrifuged to obtain serum. The levels of AST and ALT were quantified using Mouse AST and ALT ELISA kits (Abcam, Cambridge, UK) to assess hepatotoxicity of extracts A and B.

### 4.8. Statistical Analyses

Each experiment was performed a minimum of three times. Results are presented as mean ± standard deviation and analyzed using ANOVA followed by Tukey’s multiple comparison test (Sigma-plot v10.0). A *p*-value of less than 0.05 was considered statistically significant.

## 5. Conclusions

This study presents the initial evidence that RWE stimulates hair growth by enhancing factors that regulate hair growth and suppressing those that inhibit it. The treatment of RWE led to a significant increase in the expressions of VEGF and EGF, as well as the level of IGF-1, and a decrease in the expression of DKK-1 and IL-6. RWE increases the phosphorylation of ERK and AKT, and then GSK-3β phosphorylation is enhanced. Eventually, RWE promotes hair follicle growth through ERK, AKT, and the Wnt/β-catenin pathway. RWE stimulated hair growth, accelerated the onset of the anagen phase, increased the number of hair follicles, and enhanced the ratio of the anagen phase to the telogen phase in mice. Furthermore, RWE increased hair thicknesses in mouse hair. Taken together, our findings suggest that RWE could be a potential functional food for improving hair loss and hair health and a useful therapeutic agent for the treatment of alopecia and related diseases.

## Figures and Tables

**Figure 1 ijms-25-11362-f001:**
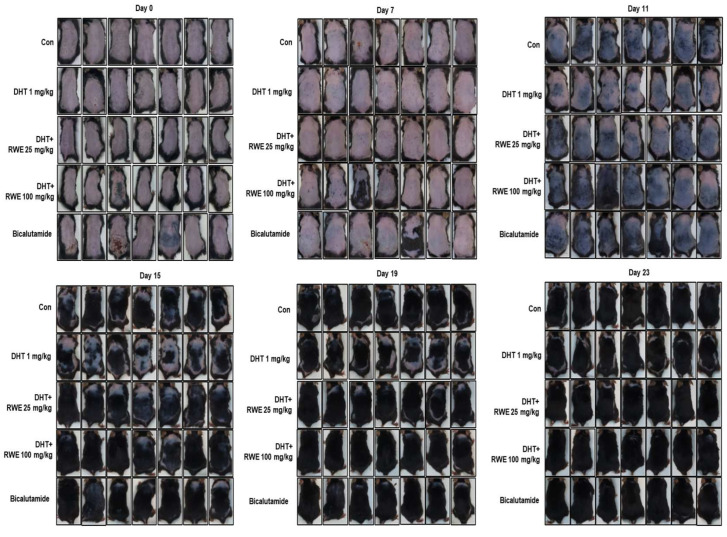
Effects of RWE on hair growth stimulatory and inhibitory factors in DHT-induced mice. Photographs of the dorsal skin of mice in each experimental group, taken on days 0, 7, 11, 15, 19, and 23. The images represent the control group (CON), DHT-induced hair loss group (DHT), and groups treated with varying doses of RWE extracts, and bicalutamide.

**Figure 2 ijms-25-11362-f002:**
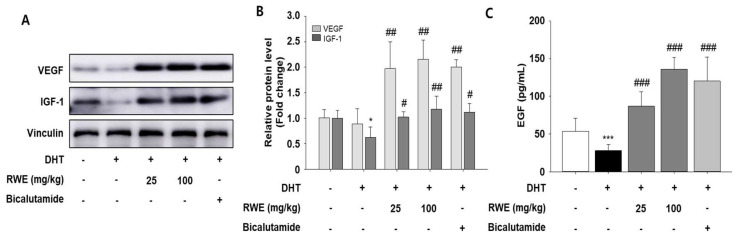
Effects of RWE on hair growth stimulatory factors in DHT-induced mice. Protein expression of VEGF and IGF-1 were determined via Western blotting (**A**). Equal quantities of total proteins were effectively analyzed using SDS-PAGE. Blot quantitative analysis (**B**). Relative protein band densities adjusted to the β-actin level were used for visualizing the densitometry data. EGF levels measured by ELISA in serum (**C**). Data are presented as mean ± SD. The data were evaluated using one-way ANOVA, with subsequent analysis performed using Tukey’s multiple comparison test. * *p* < 0.05 and *** *p* < 0.001 vs. control group; ^#^
*p* < 0.05, ^##^
*p* < 0.01, and ^###^
*p* < 0.001 vs. DHT group.

**Figure 3 ijms-25-11362-f003:**
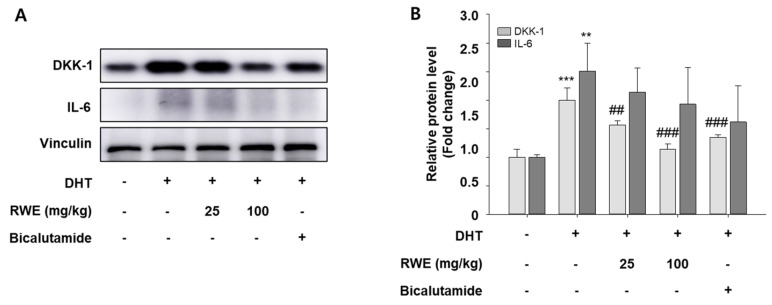
Effects of RWE on hair growth inhibitory factors in DHT-induced mice. Protein expression of DKK-1 and IL-6 were determined via Western blotting (**A**). Equal quantities of total proteins were effectively analyzed using SDS-PAGE. Blot quantitative analysis (**B**). Data are presented as mean ± SD. The data were evaluated using one-way ANOVA, with subsequent analysis performed using Tukey’s multiple comparison test. ** *p* < 0.01 and *** *p* < 0.001 vs. control group; ^##^
*p* < 0.01 and ^###^
*p* < 0.001 vs. DHT group.

**Figure 4 ijms-25-11362-f004:**
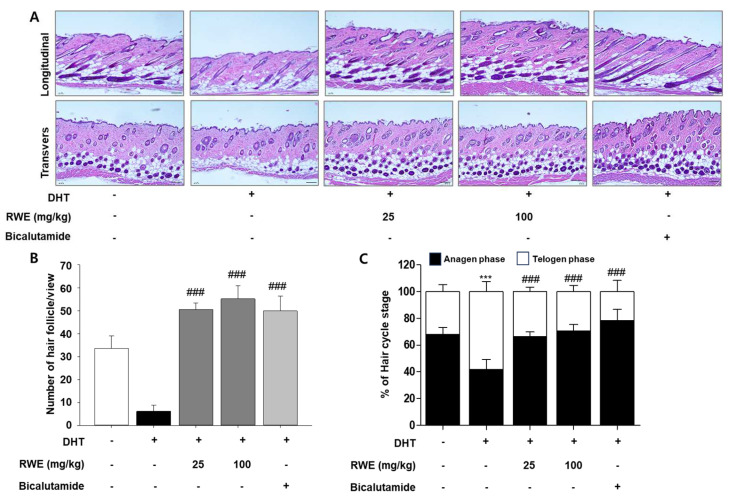
Histological examinations of the hair follicles in DHT-induced mice. Sample histopathological images of hematoxylin and eosin (H&E) staining sections of dorsal skin in longitudinal (**upper panel**) and transvers (**lower panel**) sections are shown (**A**). Scale bar represents 100 μm. Based on histological examination of the dorsal skin of mice, the number of hair follicles treated with RWE extracts were measured on day 23 (**B**). Hair growth pattern (anagen/telogen ratio) in mice (**C**). The epidermal thickness was assessed at five randomly chosen locations per slide. Data are presented as mean ± SD. The data were evaluated using one-way ANOVA, with subsequent analysis performed using Tukey’s multiple comparison test. *** *p* < 0.001 vs. control group; ^###^
*p* < 0.001 vs. DHT group.

**Figure 5 ijms-25-11362-f005:**
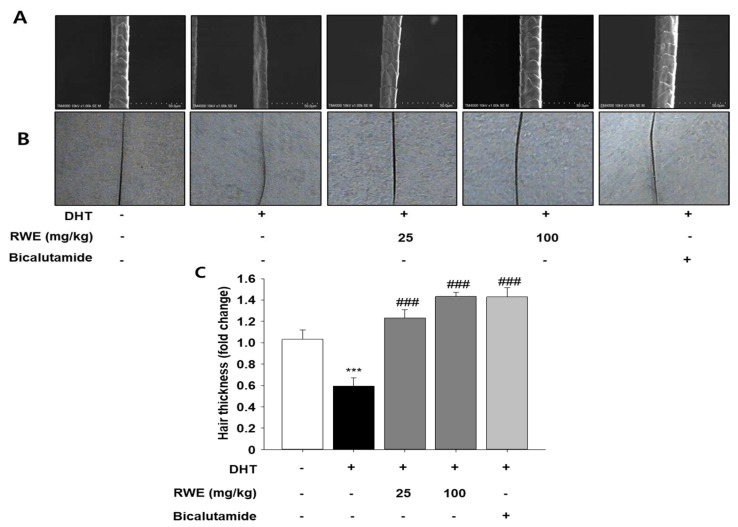
Effects of RWE on hair thickness in DHT-induced mice. Scanning electron microscopy (SEM) (**A**) and light microscopy images (**B**) of hair shafts, showing structural integrity and surface texture. Hair shaft thickness measurements (**C**). The data were evaluated using one-way ANOVA, with subsequent analysis performed using Tukey’s multiple comparison test. *** *p* < 0.001 vs. control group; ^###^
*p* < 0.001 vs. DHT group.

**Figure 6 ijms-25-11362-f006:**
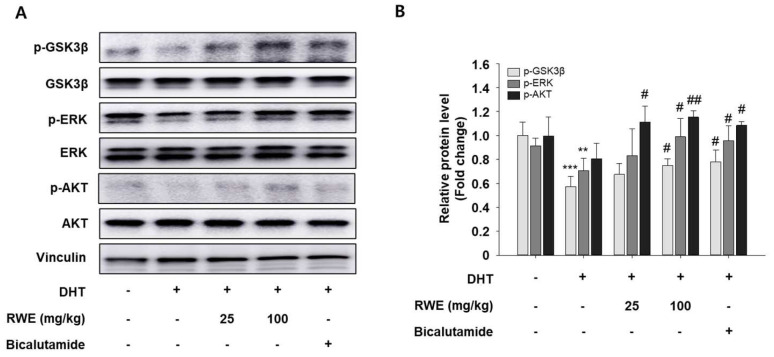
Effects of RWE on protein expression related with hair growth regulatory pathways in skin tissue of C57BL/6 mice. Protein expression of p-GSK-3β, pERK, and p-AKT were determined via Western blotting (**A**). Equal quantities of total proteins were effectively analyzed using SDS-PAGE. Blot quantitative analysis (**B**). Relative protein band densities adjusted to the β-actin level were used for visualizing the densitometry data. Data are presented as mean ± SD. The data were evaluated using one-way ANOVA, with subsequent analysis performed using Tukey’s multiple comparison test. ** *p* < 0.01 and *** *p* < 0.001 vs. control group; ^#^
*p* < 0.05 and ^##^
*p* < 0.01 vs. DHT group.

**Table 1 ijms-25-11362-t001:** Experimental group.

Group	Induction	Treatment Administration
CON	-	Saline
DHT	DHT 1 mg/kg	Saline
RWE 25 mg/kg	DHT 1 mg/kg	RWE 25 mg/kg
RWE 100 mg/kg	DHT 1 mg/kg	RWE 100 mg/kg
Bicalutamide	DHT 1 mg/kg	Bicalutamide 0.5 mg/kg

## Data Availability

The data presented in this study are available upon request.

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
