# Peer review of "Hair Growth Effect and the Mechanisms of Rosa rugosa Extract in DHT-Induced Alopecia Mice Model"

_ijms, 2024, doi:10.3390/ijms252111362_

Round 1

Reviewer 1 Report

Comments and Suggestions for Authors

Dear editor,

The article Hair Growth Effect and Their Mechanisms of Rosa rugosa Extract in DHT-Induced Alopecia Mice Model, submitted for editorial review in IJMS by Ha-Rim Kim et al, addresses an issue of medical importance for which there is no effective solution so it is certainly an important issue for the readers of the journal.

Although the proposal is interesting, I have several suggestions it is important for the authors to address before considering publication in IJMS.

Comments

Introduction.- I recommend going deeper into the biology of the hair cycle (anagen, catagen, telogen). Also, some information regarding molecular pathways associated with hair growth modulated by growth factors or those molecular pathways that suppress the hair phases.

No mention about the status of nutraceuticals for alopecia was made in the manuscript, nor their impact or limitations, to better place this study in the context of current research.

The antecedents (references 13 to 18) do not agree with the proposed nutraceutical approach for Rosa rugosa. For example, reference 14 focuses on the evaluation of the protective and regenerative properties of artichoke leaf extract on plasma and hepatic oxidative stress, and not on hair growth. Similarly, in reference fifteen, red ginseng oil was used topically, and in reference 16, a cream with nanostructured lipid carriers (NLCs) with vitamin E was used, also for topical application. None of these studies focused on oral administration with a nutraceutical approach as proposed in this work.

Methodology.-

1.- In the Western blot assays, the dilutions of the antibodies and the company from which the reagents were purchased are not indicated.

2.- Please detail how they evaluated the phases of the capillary cycle in the methodology.

3.- Please describe the criteria to identify the anagen and telogen phases.

4.- The catagen phase was not considered in the analysis, which is relevant for the exhaustive study of the capillary cycle.

Results.-

5.- It is observed in the images that the depilation of the animals was not homogeneous, which could generate variations in the results, not attributable to the treatments, due the heterogeneity in the degree of depilation.

6.- In some mice, skin damage is observed, which could influence the results due to the underlying inflammatory process.

6.- On the other hand, as the authors themselves indicated in the introduction, minoxidil and finasteride are the only treatments approved by the FDA (line 61). However, neither of them is used as a positive treatment control in the study. Since they employ a murine model based on the androgen pathway, it would be appropriate to use finasteride as a positive control, or even minoxidil, because it is known to modulate androgen receptor activity [1, 2].

7.- Although the use of RWE shows an improvement in hair growth between days 11 and 19, it lacks a quantitative analysis. I strongly suggest performing pigmentation densitometry analysis or measuring hair length on different days, which would complement Figure 1.

8.- The authors evaluate EGF in serum and observed an increase with RWE, but there is no mention of how they ensure that this growth factor directly influences hair follicles or their hair microenvironment, since it was not evaluated in cutaneous samples.

9.- VEGF and IGF-1 were assessed by Western blot; while DKK-1 and IL-6 were measured in skin samples by qRT-PCR, and EGF was measured in serum by ELISA. Could the authors explain the reasons for these methodological differences ?. The heterogeneity in the techniques employed shows inconsistency. All measurements should be performed on skin tissue and using same techniques such as Western blot or RT-PCR depending on what the authors are looking for, so that comparisons are clear and effective.

10.- In Figure 2D, they mention that RWE 100 mg/kg decreases DKK-1 and IL-6 gene expression, but only the reduction of IL-6 is statistically significant, which is inconsistent and generates confusion in the interpretation of the results. It would be important to clarify this point.

Figures.-

11.- In Figure 3, although they mention the magnification used, it is necessary to include a Scale Bar to provide an accurate size reference and allow reliable measurements.

12.- In the caption of Figure 3 it is mentioned that the epidermal thickness was assessed at five randomly chosen locations per slide, but they were not shown in the figure nor described in the manuscript.

13.- It is also suggested to enrich Figure 3 with additional measurements, such as the diameter and length of the follicles, as well as the thickness of the epidermis, dermis, or hypodermis, which are related to the phases of the hair cycle.

14.- The species name should be presented in italics, and it is necessary to homogenize both its use in italics and its abbreviated form throughout the manuscript.

15.- Finally, there is inconsistency in the use of upper and lower case for “Hematoxylin and Eosin (H&E)”, which should be corrected. Line 313, and 163.

Final suggestions.

We strongly suggest evaluating DKK-1 and IL-6 expression by Western blot, as the proteins are the effector proteins, and even if the genes are transcribed, this does not necessarily guarantee that the proteins are increased. We also propose to analyze other key proteins such as androgen receptor, β-catenin, WNT, and AKT, which participate in proliferation pathways and the androgenic pathway and could provide a more complete picture of the molecular mechanisms evaluated.

Comments on the Quality of English Language

Some details of grammar and typing errors

Reviewer 2 Report

Comments and Suggestions for Authors

This manuscript only provides the phenomenon. This manuscript did not have any underlying mechanism for the effects of RWE on hair growth. 

Considering the recent quality of IJMS, the quality of this manuscript was so low. 

Here are comments

1. The authors should provide the data about the underlying mechanisms such as Wnt signaling or ERK/Akt, etc 

2. The authors provide only 5 Figures. Figure 5 can be even the supplementary figure, because this figure is not related to hair growth. 

3. The hair cycle was not synchronized in the animal model Figure 1. The authors had better omit a few animals that were not synchronized on days 0 and 7. 

3.  RWE should be treated in DHT-induced mice to compare only DHT mice. 

Thus,  all labeling in Figures was wrong. That is, RWE and Bicalutamide should be labeled with DHT. 

4. RWE (100 mg/kg) did not affect the expression of DKK1 in  Figure 2D. The authors should mention the statistical importance of RWE (100 mg/kg) or Bicalutamide treatment compared to the control or only the DHT group. 

5. The results 2.4 and 2.5 were too short and only written in descriptive words.  

6. Despite important well-known scientific facts, one sentence just mentions only one paper in the Discussion part. 

Round 2

Reviewer 1 Report

Comments and Suggestions for Authors

The authors have made significant progress in improving the manuscript; however, several responses and points discussed lack information or clarity. Before to be consider for publication, minor details of the manuscript need to be refined.

1.        In response 5, the authors cited two articles, references 48 and 49, on lines 345 and 346, indicating that they followed the same procedure for evaluating the phases of the follicular cycle. However, they did not justify in their study why the Cartagena phase was excluded from their analysis. Please include the reason for excluding this phase in the manuscript.

2.        The authors mention that minoxidil is not a DHT blocker, which led to its dismissal as a positive control in their study. However, bicalutamide is also not a direct blocker of dihydrotestosterone (DHT); instead, it is an antagonist of androgen receptors. It works by preventing androgens, such as testosterone and DHT, from binding to their cellular receptors, thereby inhibiting their biological effects.

Recent studies have shown that minoxidil not only indirectly affects DHT synthesis by acting on key enzymes involved in its synthesis pathway, such as 17-alpha-hydroxylase/17,20-lyase steroid (CYP17A1) and aromatase (CYP19A1), but more importantly, it has been found to inhibit the expression of the androgen receptor, thus promoting hair growth along with enhanced vascularization [1,2]. Additionally, a crystallographic structure available on the PDB platform demonstrates that minoxidil can bind to the androgen receptor (Figure 1).

We kindly request that you justify your choice of bicalutamide over minoxidil in your study, given that the latter is widely accepted, commercially available, and FDA-approved. Furthermore, minoxidil may apparently have been a better positive control than bicalutamide.

1.        Shen, Y., Zhu, Y., Zhang, L., Sun, J., Xie, B., Zhang, H., & Song, X. (2023). New target for minoxidil in the treatment of androgenetic alopecia. Drug Design, Development and Therapy, 17, 2537–2547. https://doi.org/10.2147/DDDT.S427612

2.        Hsu, C.-L., Liu, J.-S., Lin, A.-C., Yang, C.-H., Chung, W.-H., & Wu, W.-G. (2014). Minoxidil may suppress androgen receptor-related functions. Oncotarget, 5(8), 2187–2197. https://doi.org/10.18632/oncotarget.1886

Figure 1. Crystal structure of the androgen receptor ligand binding domain in complex with minoxidil.

3.        In the manuscript, the authors mention that they evaluated gene expression, but in fact, they assessed protein expression through Western blot. Please change "gene expression" to "protein expression" in lines 30, 33, 128, 130, 133, 145, 147, 195, 197, 204, and 308.

The authors added the information regarding the company from which they acquired the primary antibodies, along with the dilution used. However, they need to include this information for the secondary antibody (lines 361 and 363). Additionally, the methodology section does not mention how they evaluated EGF levels. We understand that it was measured from serum, as stated in line 141 of the manuscript, but we request that this information also be included in the methodology section, along with the details of the ELISA ki

Reviewer 2 Report

Comments and Suggestions for Authors

Dear Authors

Authors response to the reviewer’s feedback very well.

Authos provide more meaningful data. 

This revised manuscript was highly improved.

Here are minor comment.

1. There was no reference in the line from 245 to 248,

Author Response

Thank you for your valuable comment.

We inserted the references in the revised MS.
